# Evolution and stabilization of subnanometric metal species in confined space by in situ TEM

Lichen Liu[1], Dmitri N. Zakharov[2], Raul Arenal [3,4], Patricia Concepcion[1], Eric A. Stach[2] & Avelino Corma[1]

Understanding the behavior and dynamic structural transformation of subnanometric metal species under reaction conditions will be helpful for understanding catalytic phenomena and for developing more efficient and stable catalysts based on single atoms and clusters. In this work, the evolution and stabilization of subnanometric Pt species confined in MCM-22 zeolite has been studied by in situ transmission electron microscopy (TEM). By correlating the results from in situ TEM studies and the results obtained in a continuous fix-bed reactor, it has been possible to delimitate the factors that control the dynamic agglomeration and redispersion behavior of metal species under reaction conditions. The dynamic reversible transformation between atomically dispersed Pt species and clusters/nanoparticles during CO oxidation at different temperatures has been elucidated. It has also been confirmed that subnanometric Pt clusters can be stabilized in MCM-22 crystallites during NO reduction with CO and $H_2$.

[1] Instituto de Tecnología Química, Universitat Politècnica de València-Consejo Superior de Investigaciones Científicas (UPV-CSIC), Av. de los Naranjos s/n, 46022 Valencia, Spain. [2] Center for Functional Nanomaterials, Brookhaven National Laboratory, Upton, New York 11973, USA. [3] Laboratorio de Microscopias Avanzadas, Instituto de Nanociencia de Aragon, Universidad de Zaragoza, Mariano Esquillor Edificio I+D, 50018 Zaragoza, Spain. [4] ARAID Foundation, 50018 Zaragoza, Spain. Correspondence and requests for materials should be addressed to A.C. (email: acorma@itq.upv.es)

Subnanometric metal catalysts (including single-atom metal species and metal clusters) have attracted great attention in material science in general and more specifically in the heterogeneous catalysis community, due to their unique properties and behaviors compared with conventional nanoparticulate catalysts[1,2]. Although it is already possible to prepare supported single atoms and metal clusters of a few atoms, stabilization of subnanometric metal catalysts under reaction conditions is a key issue and can limit the practical applications of cluster catalysts for industrial processes[3]. Subnanometric metal species can be stabilized on solid supports and their catalytic properties can be tuned by metal-support interaction[4,5]. However, under reaction conditions, the agglomeration of subnanometric metal species into nanoparticles are still difficult to avoid even when they are supported on solid carriers[6–8].

It has been presented that it is possible to generate well-defined single atoms or metal clusters in zeolites crystallites by introduction of mononuclear or multinuclear organometallic compounds into zeolites[9]. However, these subnanometric metal species are usually not stable under reaction conditions, and even under mild conditions (for instance, hydrogenation of cyclohexene at ca. 70 °C) mononuclear Ir species will migrate and stepwise agglomerate into Ir clusters and nanoparticles with reaction time[10]. Therefore, the factors that affect the stability of subnanometric species confined in zeolites need to be addressed for practical catalytic applications.

Recently, we reported a new strategy to directly generate subnanometric Pt species (Pt single atoms and clusters) in the supercages and internal cavities of MCM-22 zeolite[11]. It was found that, when Pt loading was kept ca. 0.1 wt% or lower, Pt species can show exceptional stability under consecutive oxidation–reduction treatments at high temperature (> 550 °C). As lower loadings can be a limitation for some practical applications, it is then necessary to improve the loading of Pt in MCM-22 while preserving the dispersion and stability of the subnanometric Pt species.

The fast development of in situ transmission electron microscopy (TEM) techniques in recent years has enabled researchers to study the dynamic structural evolution of nanoparticles under different atmospheres[12]. In this way, the dynamic changes in the shape of metal nanoparticles[13], oscillatory behavior of metal nanoparticles[14], reconstruction of bimetallic catalysts[15], and the interaction between metal surface and substrate molecules have already been visualized by in situ TEM[16]. However, in previous reports, the working catalysts are usually large nanoparticles (> 5 nm), which are far from most interesting subnanometric particles. Indeed, owing to limitations associated with the instrument resolution and the stability of materials under electron beam, it is quite difficult to follow the dynamic structural evolution of subnanometric metal species under reaction conditions, especially at high temperature (> 600 °C)[17].

To directly observe and obtain more detailed information on the evolution process of subnanometric metal species under redox and reaction conditions, we report herein an in situ TEM study with Pt@MCM-22 catalyst with well-defined metal species confined in the zeolite structure. Thanks to the unique structure of Pt@MCM-22 catalyst, it has been possible to have direct observations on the dynamic structural transformation of subnanometric Pt species under oxidation-reduction treatments and during CO + O$_2$, NO + CO, and NO + H$_2$ reactions. Then, by correlating the results from in situ TEM studies and the results obtained in a continuous fix-bed reactor, the factors that control the dynamic agglomeration and redispersion process, and the stability of subnanometric metal clusters under reaction conditions at high temperature have been delimitated. With a better understanding on those processes, we have been able to prepare Pt@MCM-22 catalyst with three times higher loading of Pt (ca. 0.3 wt%) within the subnanometric regime. It has also been confirmed that subnanometric Pt clusters can be stabilized by this particular zeolite structure at very high temperature (> 800 °C) during NO reduction with CO and H$_2$.

## Results

**Structural evolution of Pt species under redox conditions.** A purely siliceous Pt@MCM-22 sample with 0.17 wt% Pt was synthesized following a modification of our previously reported preparation method (see Methods for experimental details)[11]. After the calcination in air at 550 °C, the 0.17%Pt@MCM-22 sample is mainly consisted of highly dispersed Pt species in MCM-22 zeolite crystallites. As shown in Supplementary Fig. 1, Pt species cannot be observed in many areas due to the low contrast of atomically dispersed Pt species. Meanwhile, in some other areas, subnanometric Pt clusters as well as very few Pt nanoparticles can still be observed. With the help of high-resolution scanning TEM (HRSTEM), Pt single atoms as well as subnanometric Pt clusters in the 0.17%Pt@MCM-22 sample can be observed (see Supplementary Fig. 1 and Supplementary Fig. 2). Thus, the pristine 0.17%Pt@MCM-22 sample contains a mixture of Pt single atoms and clusters, as well as very few Pt nanoparticles. Then, the 0.17%Pt@MCM-22 sample was studied by environmental TEM (FEI Titan 80–300 ETEM) under high-angle annular dark field-STEM (HAADF-STEM) imaging model. It should be pointed out that, to minimize the effects of beam irradiation on the sample, very low dose of electron beam was used for HAADF-STEM imaging during in situ TEM studies (see Methods for experimental details).

It has been demonstrated by in situ spectroscopic techniques that, during redox catalytic processes, metal catalysts will undergo dynamic structural transformations, which is often related with the redox properties of the metal catalysts[18,19]. Therefore, before studying the evolution behavior of subnanometric Pt species under reactions that involve oxidative and reductive reactants, the structural transformation of the metal catalyst under reduction-oxidation (H$_2$–O$_2$) conditions has been studied. To show the evolution of atomically dispersed Pt species, we have chosen an area showing only the presence of a few Pt clusters without Pt nanoparticles to perform the in situ TEM experiments. When the sample was exposed to H$_2$ (0.1 torr) at 350 °C, a few Pt clusters (below 0.5 nm) started to appear (see Fig. 1a). When the temperature was increased to 550 °C and 700 °C in H$_2$ atmosphere, more subnanometric Pt species (below 0.8 nm) appeared and the average size increased (see Fig. 1b,c). The size of subnanometric Pt clusters continued to increase up to 0.8–1 nm when the temperature was increased to 800 °C (see Fig. 1d). Notably, the size of most of the Pt species remained below 1 nm even after reduction in H$_2$ at such high temperature, indicating the exceptional stability of Pt clusters encapsulated in the MCM-22 zeolite framework. This is even more exceptional if one takes into account that this is a purely siliceous zeolite.

After the reduction treatment, H$_2$ was switched to O$_2$ to study the behavior of Pt clusters under oxidative atmosphere. As it can be seen in Fig. 1e, Pt clusters start to redisperse after exposure to O$_2$ at 550 °C, which has also been observed by in situ extended X-ray absorption fine structure spectroscopy in our recent work[20]. It can be clearly seen that the number and the size of Pt species decreased after the gas was switched from H$_2$ to O$_2$. Therefore, the redispersion behavior of Pt species in O$_2$ is valid. Taking a careful look and comparing the images recorded in O$_2$ atmosphere, it can be seen that small Pt clusters (below 0.7 nm) located in the MCM-22 crystallites seems easier to become redispersed, than those larger Pt clusters or small nanoparticles

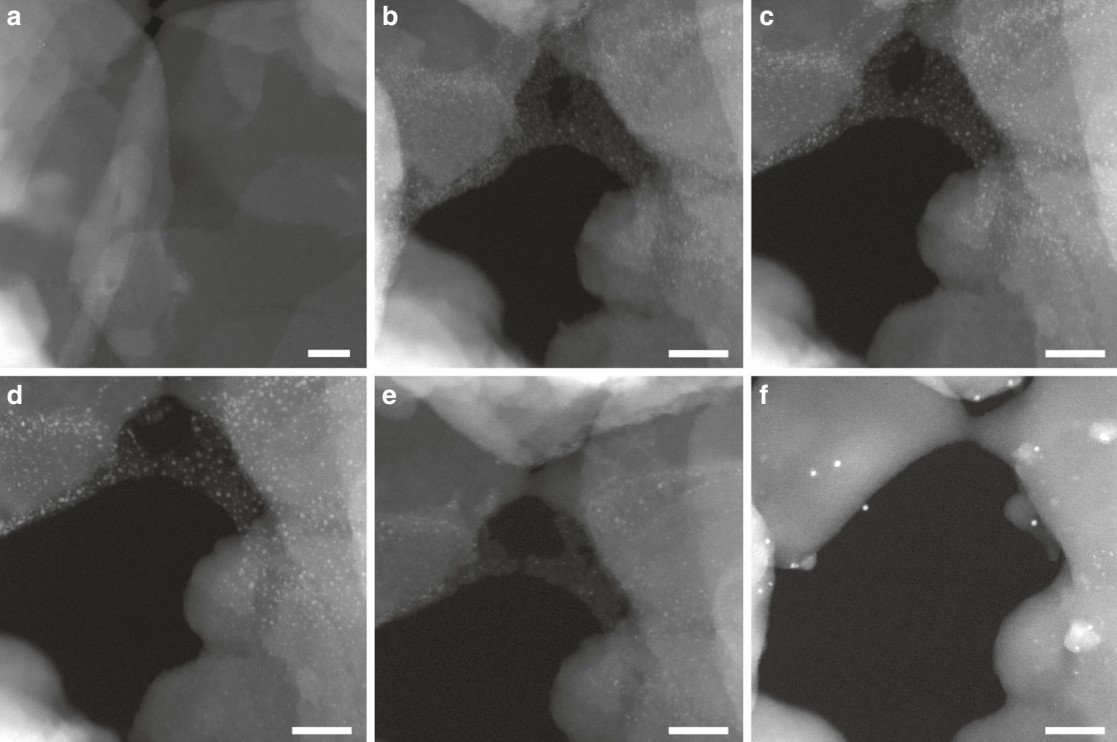

**Fig. 1** Structural evolution of 0.17%Pt@MCM-22 under redox conditions. **a** 350 °C in $H_2$ for 15 min (0.1 torr), **b** 550 °C in $H_2$ (0.1 torr) for 15 min, **c** 700 °C in $H_2$ (0.1 torr) for 15 min, **d** 800 °C in $H_2$ (0.1 torr) for 10 min, **e** 500 °C in $O_2$ (0.1 torr) for 15 min, **f** 700 °C in $O_2$ (0.1 torr) for 15 min. Scale bar in all the images in this figure: 20 nm

(~ 1 nm) located on the external surface of MCM-22. When the temperature increases to 700 °C in $O_2$, most of the Pt clusters disappear and only several small Pt nanoparticles (~ 1 nm) remain in this area (see Fig. 1f). It should be mentioned that, it seems that the MCM-22 zeolite was also damaged by the beam irradiation at high temperature. Those Pt nanoparticles may come from the sintering of Pt species located on the external surface of MCM-22 zeolite after long-time exposure to the electron beam at high temperature.

Nevertheless, all these direct observations were performed by in situ TEM. Considering the pressure gap between in situ TEM studies and more realistic catalytic reaction conditions, we have also performed ex situ TEM studies on the sample subjected to the above described oxidation–reduction treatments at atmospheric pressure. Then, when the size distributions of the Pt species were studied by TEM, it was found that highly dispersed Pt species agglomerated into clusters (< 0.5 nm) after reduction with $H_2$ at 200 °C (see Supplementary Fig. 3). Increasing the reduction temperature to 300 °C results in formation of slightly larger Pt clusters (ca. 0.5–1.0 nm), but the average size still remains in the subnanometric range (See Supplementary Fig. 4). Further increase of the temperature to 400 °C leads to formation of some Pt nanoparticles on the external surface of MCM-22 crystallites (see Supplementary Fig. 5). Notably, Pt single atoms are not observed in after reduction by $H_2$ at 400 °C. Afterwards, the redispersion of Pt nanoparticles in the reduced Pt@MCM-22 sample (by $H_2$ at 400 °C) can be achieved by another calcination in air at 550 °C (see Supplementary Fig. 6), indicating a reversible sintering and redispersion behavior of subnanometric Pt species during oxidation–reduction treatments.

It should be mentioned that, in our previous work, we have also tested the stability of subnanometric Pt species during consecutive oxidation–reduction treatments, and it was found that when the loading of Pt in Pt@MCM-22 material was ca. 0.1

wt% or lower, a large percentage of Pt species were located in the internal space of MCM-22 crystallite[11]. In that case, subnanometric Pt species and small Pt nanoparticles (~ 1 nm) can be stabilized by the zeolite structure.

Combining the results obtained from the in situ and ex situ TEM studies, the factors related with the dynamic agglomeration-redispersion behaviors are illustrated in Fig. 2. When the loading of Pt in purely siliceous MCM-22 zeolite is low (below 0.1 wt%), subnanometric Pt species can disintegrate into atomically dispersed Pt in $O_2$ under relatively mild conditions. Furthermore, as most of the Pt species are located in the internal space of MCM-22 crystallites, they are protected by the zeolite framework from agglomeration into nanoparticles during reductive treatments. However, when the loading amount increase to > 0.15 wt %, a part of the subnanometric Pt species will be located on the external surface or subsurface of MCM-22 crystallites. Therefore, in afterward reduction treatment with $H_2$ at relatively high temperature (> 400 °C), Pt species located on the external surface of subsurface will agglomerate into Pt nanoparticles (> 2 nm), as observed in Supplementary Fig. 5, whereas those Pt species located in the internal space of MCM-22 crystallites will only grow into clusters or very small nanoparticles as a consequence of the confinement in the zeolite structure.

As two of the reactions that will be investigated later by in situ TEM involve reduction of NO by CO or $H_2$, we will also study the morphological transformation of Pt species in NO atmosphere with *in situ* TEM. As it can be seen in Supplementary Fig. 7 and Supplementary Fig. 8, after exposure to NO (0.1 torr) at room temperature, the contrast of Pt species decreased, while the size increased after NO treatment. For smaller Pt species (below 1 nm), they disappear in the STEM image after the NO treatment, indicating redispersion of Pt species. Furthermore, we have also measured the size of Pt particles after NO treatment. As presented in Supplementary Fig. 9, the size of Pt particles increases slightly

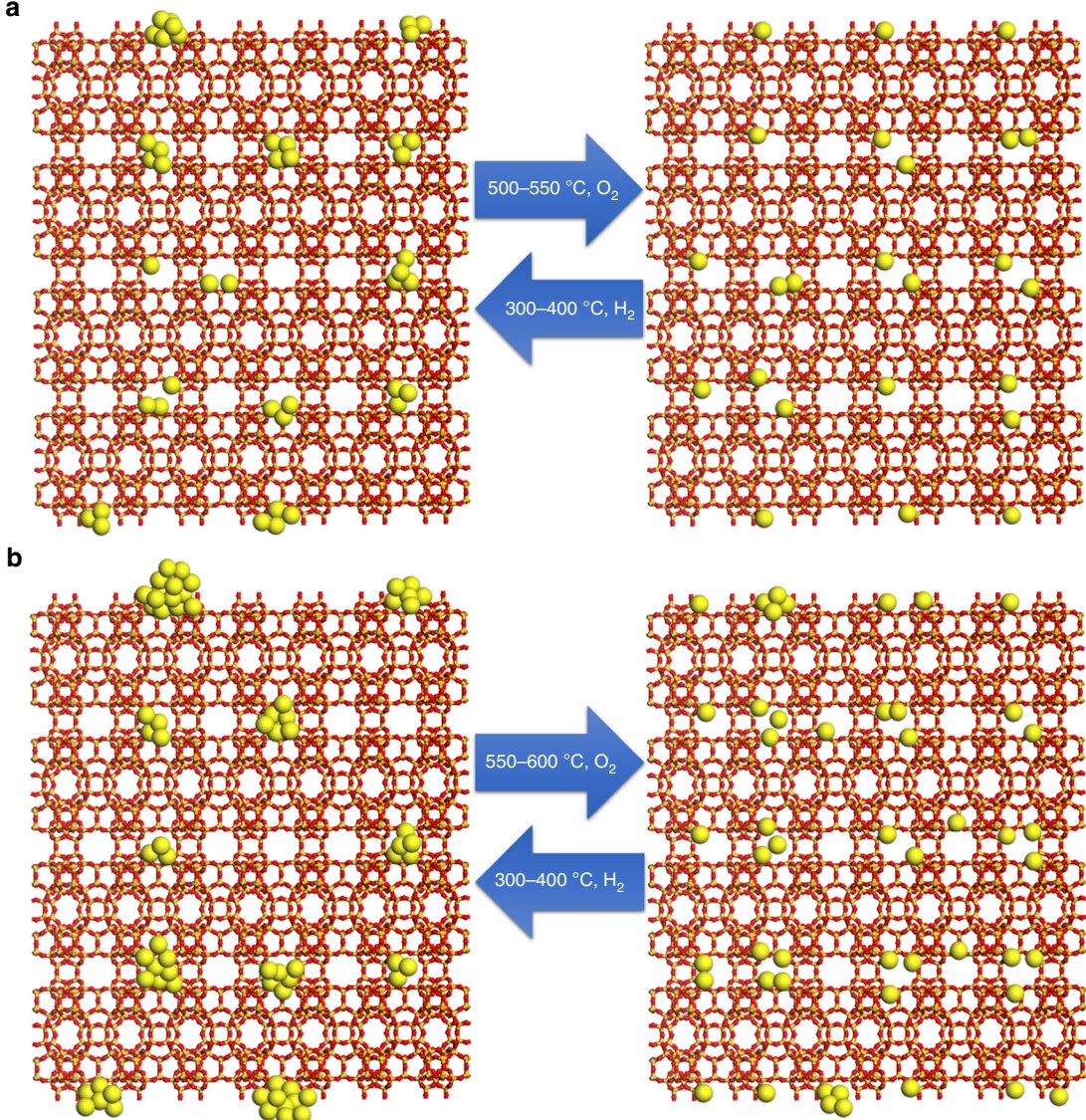

**Fig. 2** Structural evolution of Pt@MCM-22 under redox conditions. **a** In the case of Pt@MCM-22 material with low Pt loading ($\leq$ 0.1 wt%), Pt mainly exist in internal space of MCM-22 crystallites, as shown in our previous work[11]. Therefore, the redispersion of Pt can be achieved under relatively mild conditions. Subnanometric Pt clusters and small Pt nanoparticles (1–2 nm) are formed in subsequent reduction treatment by $H_2$. **b** In the case of Pt@MCM-22 material with higher loading (> 0.15 wt%), some of the Pt species will form Pt nanoparticles and locate on the external surface of MCM-22, which need treatments at higher temperature as driving force to achieve the redispersion. Subsequent reduction treatment by $H_2$ will cause the agglomeration of Pt species on the surface, giving Pt nanoparticles (> 2 nm)

after NO treatment according to the contrast profiles, implying the possible structural transformation of Pt particles[13]. However, as we have seen previously in $O_2$ atmosphere under the same conditions, Pt clusters will remain unchanged, indicating that NO is much more efficient than $O_2$ for the redispersion of Pt species. The redispersion of Pt particles in NO is probably related with the NO dissociation on Pt species, leading to the oxidative disintegration of Pt clusters and nanoparticles[21,22].

According to the above results, we have made some modifications on the preparation of Pt@MCM-22 to achieve high Pt loading with still good dispersion of subnanometric Pt species. Then, a 0.3%Pt@MCM-22 sample was prepared, and Pt nanoparticles can be seen on the surface of MCM-22 zeolite crystallites as a consequence of higher Pt loading (see Supplementary Fig. 10). At this point, treatment with NO was carried out with the aim to promote the redispersion of Pt nanoparticles

into subnanometric Pt species. As shown in Supplementary Fig. 11, subnanometric Pt clusters were observed with this 0.3% Pt@MCM-22 sample after oxidative treatment with NO at 200 °C. The high-resolution STEM image of the sample after NO treatment at 200 °C clearly demonstrate the presence of Pt clusters and single Pt atoms. However, a small amount of Pt nanoparticles was still present after NO treatment at 200 °C (see Supplementary Fig. 11a). Further increasing the temperature to 300 °C results in better redispersion of Pt nanoparticles in the sample (see Supplementary Fig. 13). It appears then that NO can also redisperse the Pt nanoparticles on the external surface of the MCM-22 crystallites. This is an important observation since it indicates that the rejuvenation of industrial catalysts containing metal nanoparticles should be better achieved by introducing NO during the treatment. More interestingly, a reduction treatment with $H_2$ at 200 °C on the NO-induced redispersed

Pt@MCM-22 sample would cause the sintering of highly dispersed Pt species into Pt clusters, as well as a small fraction of nanoparticles (see Supplementary Fig. 14). Notably, most of the Pt species are present as subnanometric Pt clusters in the reduced sample, suggesting that the NO treatment not only disintegrates the Pt nanoparticles into highly dispersed Pt species, but also led to the migration of Pt species within MCM-22 crystallites. Notice that, high-temperature calcination at 650 °C in air was necessary to fully achieve redispersion of Pt species with the 0.3%Pt@MCM-22 sample. It appears therefore that treating Pt-zeolite catalysts, or even Pt on other supports, with NO can be an effective way to achieve redispersion of Pt nanoparticles at relatively lower temperature compared with treatment with $O_2$, especially for those samples containing a higher loading of Pt species.

**Structural evolution of Pt under CO+O$_2$ conditions.** Up to now, we have studied the dynamics of the Pt nanoparticles and subnanometric species when performing in situ and ex situ treatments with $H_2$, $O_2$, or NO. Now, we will first study the stability and the evolution of subnanometric Pt species with the 0.17%Pt@MCM-22-300H$_2$ sample during CO oxidation with $O_2$ by in situ TEM. As presented in Fig. 3, highly dispersed Pt species evolved into Pt clusters when the reaction temperature for CO oxidation was increased to 100–150 °C. The agglomeration of highly dispersed Pt species should be caused by the interaction between Pt and CO[23]. Those subnanometric Pt clusters and small Pt nanoparticles (1–2 nm) remained stable when reaction temperature was increased up to 300 °C. More STEM images of the other areas of 0.17%Pt@MCM-22-300H$_2$ sample under CO + $O_2$ reaction conditions are shown in Supplementary Fig. 15. The results obtained imply that atomically dispersed Pt species can undergo in situ structural transformation under reaction conditions and the MCM-22 zeolite, even in its purely siliceous form, can stabilize subnanometric Pt clusters from sintering further into Pt nanoparticles. Surprisingly, when the reaction temperature for CO + $O_2$ reaction was increased to as high as 400 °C, Pt clusters almost remained unchanged, while the Pt nanoparticles disintegrated into Pt clusters. As subnanometric Pt clusters remain unchanged, the possibility that the disintegration is caused by the redispersion of Pt nanoparticles at high temperature due to the presence of $O_2$ under reaction conditions can be ruled out. (Fig. 4)

In a recent theoretical study, Wang et al.[24] have employed density functional theory (DFT)-based ab initio molecular dynamics simulations of Au/CeO$_2$ catalysts under CO oxidation conditions. Their work shows that single-atom Au species can be generated from Au clusters or nanoparticles as a result of interaction between Au and CO at the Au-CeO2 interface. Experimental work has also been reported on the dynamic transformation of Pt clusters or nanoparticles in CO atmosphere by spectroscopic techniques. The Pt$_{13}$ clusters confined in NaY zeolite would disintegrate into Pt$_2$(CO)$_m$ carbonyl clusters when exposed to CO[25]. According to previous experimental and theoretical work and the results presented in this work, we speculate that the disintegration of Pt nanoparticles under CO + $O_2$ reaction conditions is caused by the interaction between Pt and the CO molecules[26]. The presence of $O_2$ also contributes the stabilization of highly dispersed Pt species at high temperature. The above results show that ETEM allows to perform direct observation on the dynamic structural transformation between atomically dispersed Pt species, clusters, and nanoparticles under reaction conditions.

Then, to correlate the information presented above on the structural transformation of Pt species obtained during the in situ TEM studies, we have also conducted the catalytic studies on the

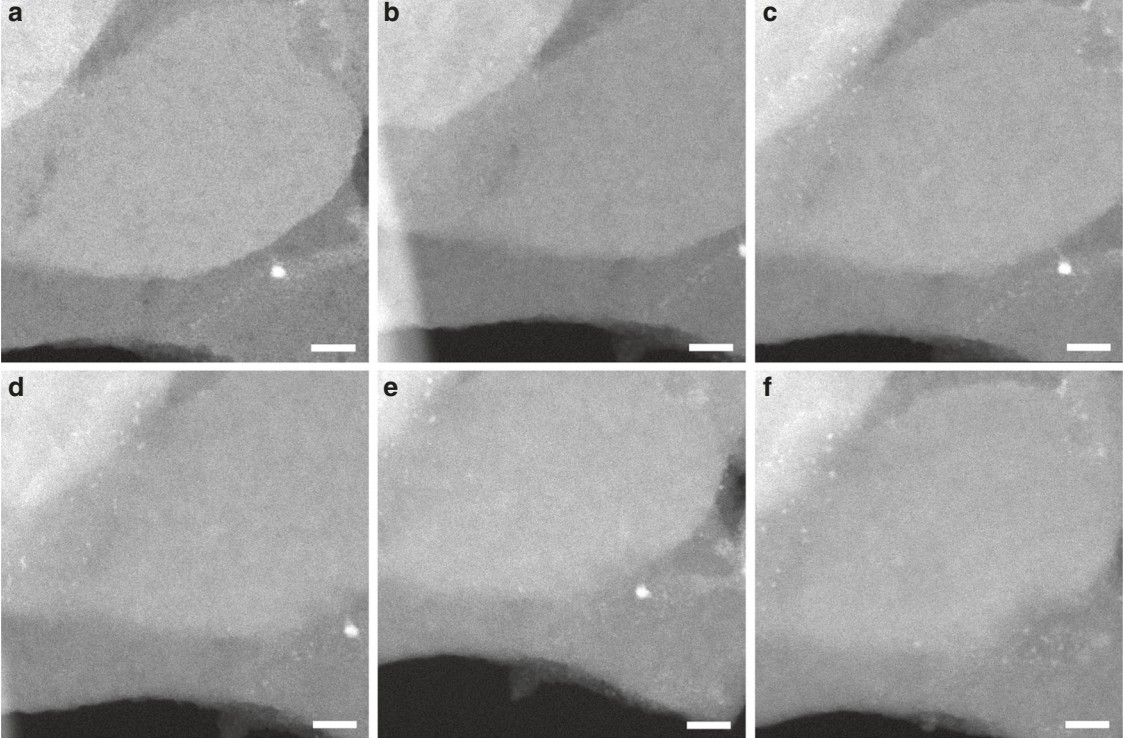

**Fig. 3** Structural evolution of 0.17%Pt@MCM-22-300H$_2$ under CO + $O_2$ conditions. During the in situ TEM experiments, the sample was treated in mixture of CO (0.2 torr) and $O_2$ (0.1 torr) for 15 min at different reaction temperature, respectively. In order to avoid the carbon deposition of CO by the electron beam, the TEM chamber was evacuated to remove CO gas after exposure of the sample to CO + $O_2$ gases. **a** Room temperature, **b** 100 °C, **c** 150 °C, **d** 200 °C, e 300 °C, **f** 400 °C. Scale bar in all the images in this figure: 10 nm

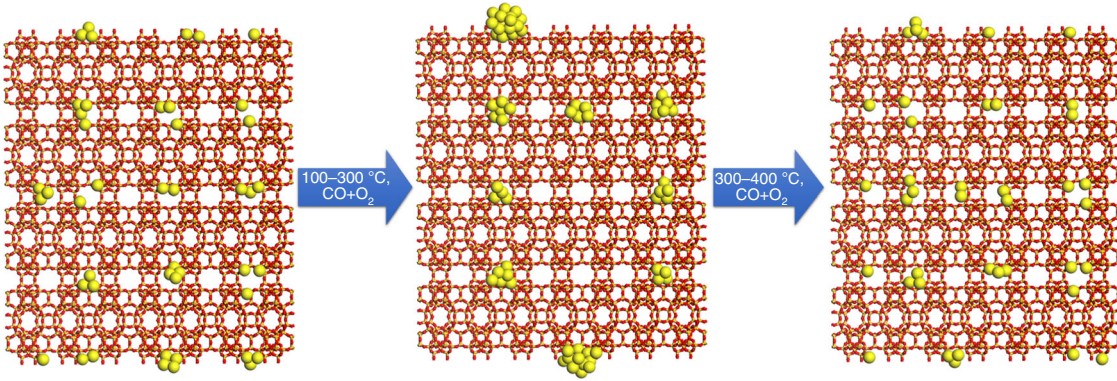

**Fig. 4** Structural evolution of 0.17%Pt@MCM-22-300H₂ under CO + O₂ conditions. At 100–200 °C, highly dispersed Pt species will agglomerate into Pt clusters or even small nanoparticles. When the reaction temperature continues to increase to higher temperature (300–400 °C), Pt nanoparticles will disintegrate and form highly dispersed Pt species and Pt clusters

0.17 wt%Pt@MCM-22 sample for CO oxidation in a fix-bed reactor. To demonstrate the transformation of highly dispersed Pt species under CO oxidation conditions, the 0.17%Pt@MCM-22 sample was first calcined in air at 550 °C to achieve the redispersion of Pt into highly dispersed Pt species in MCM-22. Therefore, in the starting catalyst, Pt species mainly exist as highly dispersed species. After performing the CO + O₂ reaction at 300 °C and achieving 100% conversion, the catalyst was studied by ex situ STEM. As shown in Supplementary Fig. 16, a large number of Pt clusters as well as a small fraction of Pt nanoparticles appear in the used catalyst, indicating the transformation of singly dispersed Pt species into Pt clusters and nanoparticles under reaction conditions, which is consistent with the results obtained with the in situ TEM experiments. After following the evolution of the Pt species and catalytic activity by in situ and ex situ STEM, we can elaborate the active sites of Pt catalysts for CO oxidation.

Furthermore, as mentioned before when discussing our in situ TEM results for CO oxidation, redispersion of Pt nanoparticles at high temperature in CO + O₂ atmosphere was observed. To confirm that phenomenon in a practical system, the 0.17% Pt@MCM-22-300H₂ sample has also been tested under CO + O₂ reaction conditions at 400 °C (full conversion of CO was achieved at 400 °C) to check the structural evolution of Pt species at high temperature. As it can be seen in Supplementary Fig. 17, the number of Pt nanoparticles decreased in the used catalyst, indicating the redispersion of Pt under CO + O₂ reaction conditions at 400 °C.

In recent years, there are some reports on the applications of supported single-atom Pt catalysts for CO oxidation reaction. For instance, Narula and colleagues[27] claimed that Pt single atoms supported on Al₂O₃ could serve as the active sites for CO oxidation. Besides, in a recent paper, Lu and colleagues[28] have prepared a Pt/CeO₂ sample containing Pt single atoms by atomic layer deposition and the catalyst was used for CO oxidation. It should be noted that only fresh catalysts were characterized in those works, whereas the evolution of the catalysts under reaction conditions were not followed. Based on these results shown above, it seems that for Pt supported on MCM-22, Pt clusters and nanoparticles are the active species for CO oxidation at 200–300 °C, while highly dispersed Pt species (single atoms and clusters) may serve as the active sites at high temperature (~ 400 °C)[29]. This phenomenon also indicates that, treating the Pt catalysts under CO + O₂ atmosphere at high temperature can be an alternative route for the transformation of Pt nanoparticles into highly dispersed Pt species.

Moreover, this strategy may also be applicable to other group VIII metal catalysts.

Nevertheless, we have also followed the evolution of Pt species by in situ infrared (IR) spectroscopy for CO + O₂ under flow conditions, and CO was used as probe molecules to study the chemical states and atomicity of Pt species after CO + O₂ reaction at different temperature[30–32]. As shown in Supplementary Fig. 18, both oxidized Pt species as well as Pt clusters were observed in the CO adsorption spectrum for the starting 0.17%Pt@MCM-22 sample and those Pt species remain almost unchanged after CO + O₂ reaction at 225 °C. When the temperature was increased to 300 °C, the amount of subnanometric Pt clusters decreased and they were transformed into oxidized Pt species, which should be caused by the sintering of Pt clusters to small Pt nanoparticles as observed by TEM. Interestingly, those agglomerated Pt nanoparticles will disintegrate into Pt clusters again after reaction at 400 °C.

Combining our results and reported works, it can be concluded that the evolution of singly dispersed metal species under reaction conditions is a critical issue when studying the catalytic properties of single-atom catalysts[33]. Special attention should be paid to the structural evolution of metal species under reaction conditions. Nevertheless, the state (particle size and chemical states) of metal species may also be dependent on the reaction conditions (atmosphere and temperature). Therefore, the possibility that different reaction mechanism and different types of active species may exist under various reaction conditions has to be considered.

**Evolution of Pt under water–gas shift conditions**. The evolution of Pt species under water–gas shift reaction condition has also been studied by in situ TEM. As shown in Supplementary Fig. 19 and Supplementary Fig. 20, the transformation of highly dispersed Pt species into Pt clusters was observed when the temperature was increased to 100 °C. Further increase of reaction temperature to 200 °C, 300 °C, and 400 °C results in further growth of Pt clusters. However, as a result of the confinement effect of MCM-22 zeolite, the Pt clusters are prevented from agglomeration into large Pt nanoparticles. Notably, Pt clusters or small Pt nanoparticles remained stable at 400 °C and the redispersion of Pt nanoparticles was not observed. This is different to the dynamic behavior of subnanometric Pt species under CO oxidation reaction conditions, wherein redispersion of Pt nanoparticles into Pt clusters and atoms occurs at 400 °C. As discussed before, the agglomeration-redispersion behavior of Pt species is strongly related with the atmosphere.

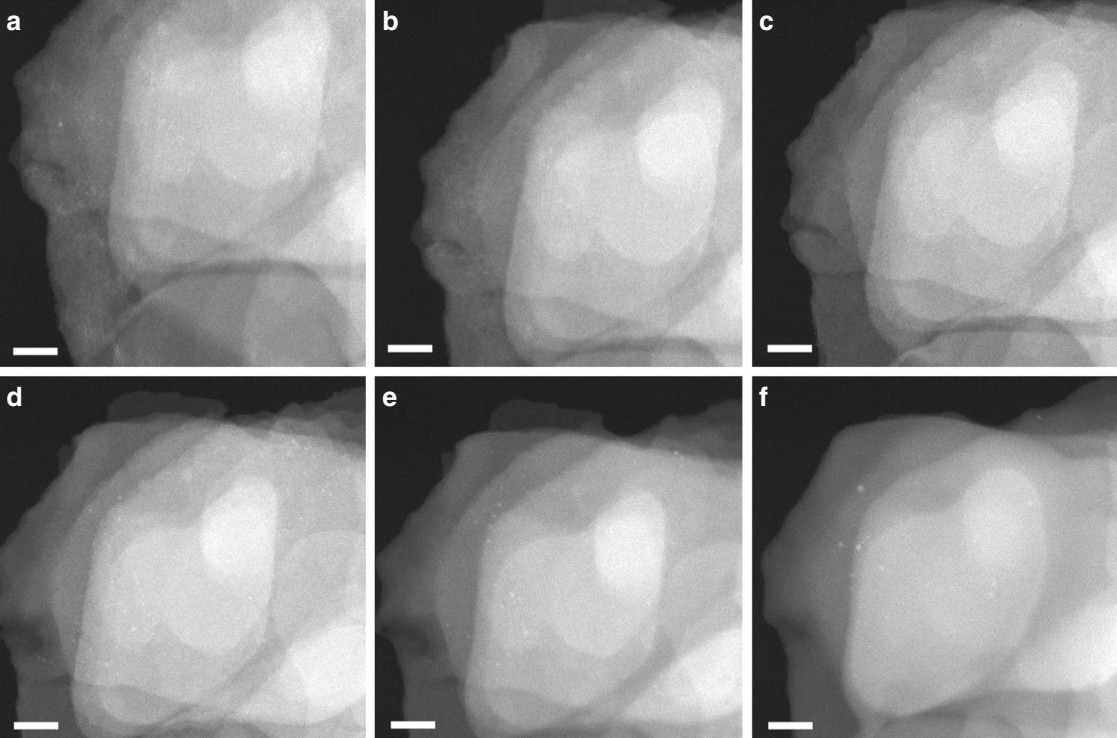

**Fig. 5** Stability test of 0.17%Pt@MCM-22-300H₂ under CO + NO conditions. During the in situ TEM experiments, the sample was treated in mixture of CO (0.1 torr) and NO (0.1 torr) for 15 min at different reaction temperature, respectively. In order to avoid the carbon deposition of CO by the electron beam, the TEM chamber was evacuated to remove CO gas after exposure of the sample to CO+NO gases. **a** 200 °C, **b** 400 °C, **c** 600 °C, **d** 800 °C, **e** 1,000 °C, **f** 1,200 °C. Scale bar in all the images in this figure: 20 nm

**Evolution of Pt under selective reduction of NO by CO or $H_2$.** In this section, selective reduction of NO by CO or $H_2$ has been chosen as model reaction for studying the behavior of the sub-nanometric Pt species in the Pt@MCM-22-300H₂ sample under the reaction conditions of elimination of NOx in exhaust gases[34–37]. As shown in Fig. 5a, subnanometric Pt species were observed at 200 °C in the presence of CO + NO. Interestingly, it appeared that when the temperature was increased from room temperature to 400 °C (see Fig. 5b), the contrast of subnanometric Pt species decreased, implying some structural transformation. Considering our abovementioned results for the dynamic agglomeration-redispersion behaviors of Pt species, it can be speculated that Pt clusters become redispersed at 200−400 °C in CO + NO atmosphere. Similar phenomenon has also been observed in another area of the sample, as shown in Supplementary Fig. 21. When the temperature was increased to 800 °C, subnanometric Pt clusters with higher contrast appeared again. Further increase of the temperature to 1,000 °C or even 1,200 °C results in the growth of the particle size of Pt clusters. Notably, most of the Pt species are still below 1 nm with a small fraction of Pt nanoparticles (~ 1 nm), indicating the excellent stability of Pt nanoclusters at high temperature when encapsulated in MCM-22 zeolite. More STEM images of the other areas of 0.17%Pt@MCM-22-300H₂ sample under CO + NO reaction conditions are shown in Supplementary Fig. 22, showing similar tendency during the process from 200 °C to 1,200 °C under reaction conditions.

The stability of subnanometric Pt species during the reaction of NO with $H_2$ has also been investigated by in situ TEM. As shown in Fig. 6, the behavior of Pt species shows a similar evolution pattern to that observed during the NO + CO reaction. More STEM images of the other areas under NO + $H_2$ reaction conditions are shown in Supplementary Fig. 23. Interestingly,

when the temperature was increased from room temperature to 200 °C, redispersion of Pt clusters to atomically dispersed Pt species was observed (see Supplementary Fig. 24), which is similar to the situation for NO + CO reaction when increasing the reaction temperature. Furthermore, at 400−600 °C, Pt mainly existed as subnanometric or atomically dispersed Pt species, with low contrast in STEM images. When the temperature was increased to 800 °C, Pt clusters as well as a small fraction of Pt nanoparticles (~ 1 nm) appeared, corresponding to structural transformation at this temperature. The evolution of Pt species in MCM-22 from room temperature to high temperature is summarized in Fig. 7.

Based on the above results obtained from in situ TEM studies, we can have a summary on the evolution of subnanometric Pt species under different reaction conditions, as presented in Fig. 8. Under reductive atmosphere (CO + $O_2$ and CO + $H_2O$), atomically dispersed Pt species will agglomerate into Pt clusters (at 100−300 °C). However, in an oxidative atmosphere (NO + $H_2$ and NO + CO), subnanometric Pt clusters will disintegrate into atomically dispersed Pt species at relative lower temperature (200−400 °C). At higher temperature, the behavior of Pt species is also strongly related with the atmosphere.

In summary, the dynamic structural transformation of encapsulated subnanometric Pt species in zeolites (including atomically dispersed Pt and Pt clusters) has been studied with in situ TEM under oxidation-reduction and reaction conditions. Comparing with conventional Pt nanoparticles, the behaviors of subnanometric Pt species is much more sensitive to the presence of reactants. Dynamic and reversible transformation between single atoms, clusters and nanoparticles has been observed under CO + $O_2$ reaction conditions at different temperature. Furthermore, their local coordination environment (in this work, this

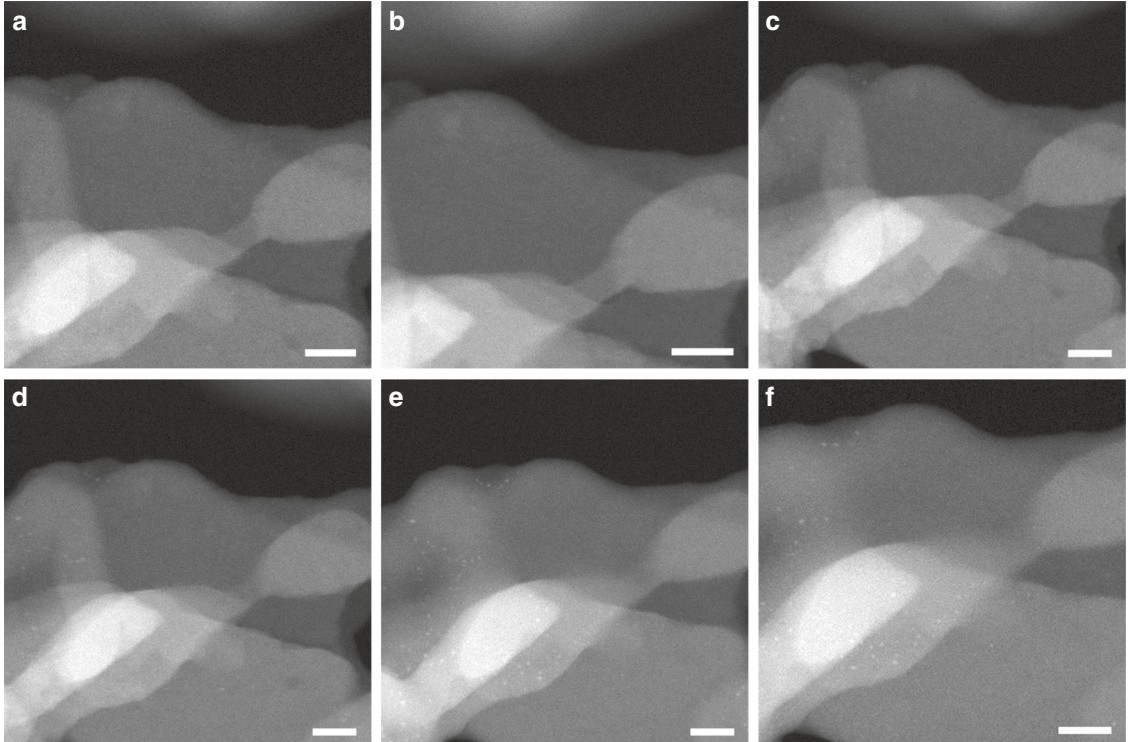

**Fig. 6** Stability test of 0.17%Pt@MCM-22-300H$_2$ under NO + H$_2$ conditions. During the in situ TEM experiments, the sample was treated in mixture of H$_2$ (0.1 torr) and NO (0.1 torr) for 15 min at different reaction temperature, respectively. The above images were recorded in the presence of NO+H$_2$ gases at different temperature. **a** 200 °C, **b** 400 °C, **c** 600 °C, **d** 800 °C, **e** 1,000 °C, **f** 1,200 °C. Scale bar in all the images in this figure: 20 nm

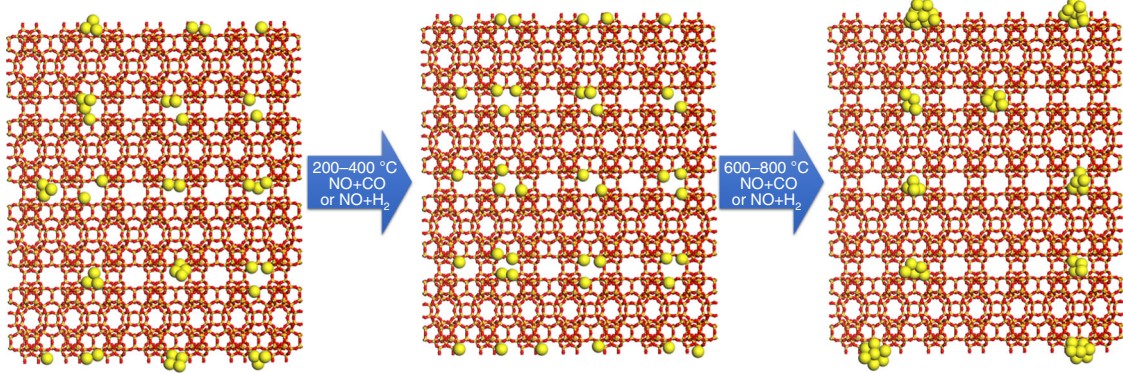

**Fig. 7** Structural evolution of Pt species under CO + NO and NO + H$_2$ conditions. At 200–400 °C, Pt clusters will disintegrate and form highly dispersed Pt species. At higher temperature (600–800 °C), highly dispersed Pt species agglomerate into Pt clusters or even small Pt nanoparticles (1–2 nm). Due to the protection effect from MCM-22 framework, Pt clusters and small nanoparticles (1–2 nm) can be stabilized at such high temperature

factor corresponds to the location of Pt species in the zeolite crystallites) also affects the behaviors of Pt species. By tuning the size and spatial distribution of Pt species in MCM-22, subnanometric Pt clusters can be stabilized under reaction conditions, even at very high temperature (> 800 °C).

Nevertheless, considering our results shown in this work obtained under different conditions, it can be a general phenomenon that subnanometric metal species will undergo dynamic structural evolution under reaction conditions[18,38]. For the same reaction, the states of metal species are dependent on the temperature and atmosphere. Therefore, when studying the catalytic properties of subnanometric metal species (single atoms and clusters), in situ or operando characterization is necessary when one tries to identify the active species for specific reaction.

Finally, by studying the evolution patterns of subnanometric metal species under reaction conditions, it can also help us to develop new methodologies (using different pretreatment conditions) for preparation of metal catalysts containing subnanometric metal species for catalytic applications.

## Methods

**Preparation of 0.17%Pt@MCM-22 sample**. The 0.17%Pt@MCM-22 sample was prepared according to our previous work, with minor modifications. First, subnanometric Pt species were prepared by reduction of Pt precursor with *N,N*-dimethylformamide (DMF). H$_2$PtCl$_6$ (110 mg) was dissolved in 120 ml DMF and then the solution was heated at 140 °C for 6 h. After being heated at 140 °C for 6 h, a yellow solution was obtained.

The subnanometric Pt species were incorporated into purely siliceous ITQ-1 by a swelling process. ITQ-1 (0.75 g), 3.0 g of H$_2$O, 15.0 g of cetyltrimethylammonium

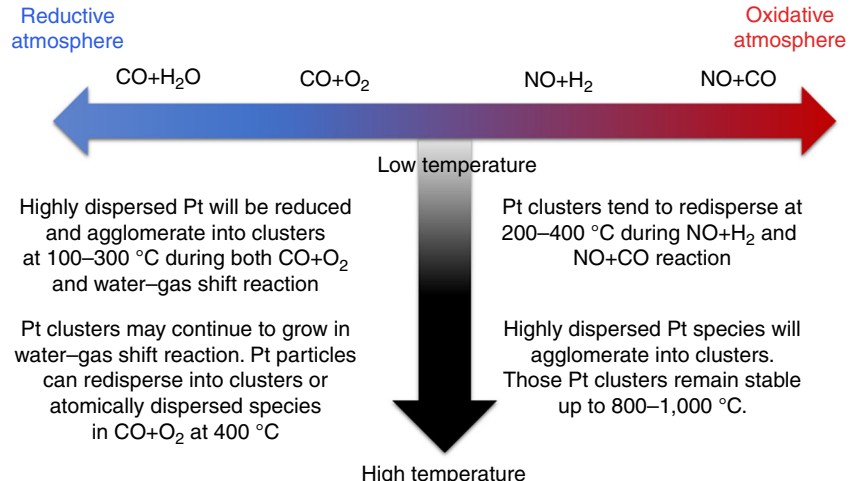

**Fig. 8** Comparison of the evolution of Pt species under different reaction conditions. The evolution of subnanometric Pt species is related with the reactants and reaction temperature

hidroxide solution (50 wt., 50% exchanged Br⁻/OH⁻), and 4.5 g of tetrapropylammonium hydroxide solution (40 wt., 50% exchanged Br⁻/OH⁻) were mixed together with 15 ml of the DMF solution containing subnanometric Pt species at room temperature under vigorous stirring. The resultant suspension was kept at 52 °C under stirring vigorously for 16 h. After the swelling process, the solid was recovered by centrifugation at 6,000 r.p.m. and then washed with distilled water. Finally, the solid product was dried in an oven at 60 °C in air. The dried solid product containing subnanometric Pt species was calcined in flow air to remove the organic templates and surfactants at 550 °C for 4 h with a ramp rate of 2 °C min⁻¹ from room temperature to 550 °C. After removing the organic templates and surfactants, the solid sample was calcined again in flow air at 550 °C for 4 h with a ramp rate of 2 °C min⁻¹ from room temperature to 550 °C, resulting in the formation of 0.17%Pt@MCM-22 sample. The loading of Pt was measured by inductively coupled plasma.

0.17%Pt@MCM-22-300H₂ was prepared by reducing 0.17%Pt@MCM-22 sample with H₂ at 300 °C for 30 min with a ramp rate of 5 °C min⁻¹ from room temperature to 300 °C.

NO gas (4% of NO in N₂) was used to oxidize the Pt species in Pt@MCM-22 sample to achieve redispersion of Pt nanoparticles in the Pt@MCM-22 with 0.3 wt% of Pt.

**Characterizations**. Samples for ex situ electron microscopy studies were prepared by dropping the suspension of Pt@MCM-22 or other materials using ethanol as the solvent directly onto holey-carbon coated Cu grids. The measurements were performed in a JEOL 2100 F microscope operating at 200 kV both in TEM and STEM modes. STEM images were obtained using a HAADF, which allows Z-contrast imaging. High-resolution STEM measurement was performed on FET Titan low-base microscope at 300 kV equipped with a Cs probe corrector, a monochromator and an ultrabright X-FEG electron source. The convergence angle was 25 mrad and the inner and outer angles for HAADF imaging were 70 and 200 mrad, respectively. The typical probe current was set to 2 pA under the HRSTEM imaging conditions. Control experiments have been performed and it was found that, under our experimental conditions, the subnanometric Pt species remain stable under the electron beam in the first several acquisitions.

In situ electron microscopy experiments were performed using a Titan 80–300 Environmental Transmission Electron Microscope at the Centre for Functional Nanomaterials, Brookhaven National Laboratory. The reaction conditions and experimental procedures were presented in the figure captions for different reactions. The in situ TEM experiments were performed at 300 kV. The beam current is 0.418 nA and the dose rate is 176 e A⁻² s⁻¹. The convergence angle, the inner angle and the outer angle is 10, 25 and 51 mrad, respectively. TEM chips from DENS solution with through holes were used in all the in situ TEM experiments.

In situ IR experiments. IR spectra were recorded with a Bruker spectrometer, Vertex 70, using a DTGS detector and acquiring at 4 cm⁻¹ resolution. An IR cell allowing in situ treatments in controlled atmospheres and temperatures from 25 °C to 500 °C was connected to a vacuum system with gas dosing facility. For IR studies, the samples were pressed into self-supported wafers and treated at 150 °C in vacuum (10⁻⁴ mbar) for 1 h. Afterwards the sample has been exposed to a flow of 1.6 ml min⁻¹ of CO and 0.8 ml min⁻¹ of O₂ (CO/O₂ = 2/1 molar ratio) for 1 h and at different temperatures, 225 °C, 300 °C, and 400 °C. Each temperature corresponded to an independent experiment. After being kept at the corresponding temperature, the sample was cooled down to room temperature, and then

evacuated at 10⁻⁴ mbar and followed by CO dosing at increasing pressure (0.4–2.5 mbar). IR spectra were recorded after each dosage. An additional CO adsorption experiment of the fresh catalyst was done on the Pt@MCM-22 sample without being exposed to CO + O₂ reaction.

**Catalytic measurement**. The CO oxidation was performed in a fix-bed reactor. Solid catalyst (120 mg) was used for each test. The feed gas was 2% of CO and 1% of O₂ in He. The total flow was 85 ml min⁻¹. The product was analyzed by gas chromatograph with thermal conductivity detector.

**Data availability**. The data that support the findings of this study are available from the corresponding author upon request.

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

## Acknowledgements

This work has been supported by the European Union through the European Research Council (grant ERC-AdG-2014-671093, SynCatMatch) and the Spanish government through the "Severo Ochoa Program" (SEV-2016-0683). L.L. thanks ITQ for providing a contract. We also thank Microscopy Service of UPV for the TEM and STEM measurements. The in situ TEM experiments were performed in the Center for Functional Nanomaterials, which is a US DOE Office of Science User Facility, at Brookhaven National Laboratory under contract number DESC0012704. The HAADF-HRSTEM studies have been conducted in the Laboratorio de Microscopias Avanzadas (LMA) at the Instituto de Nanociencia de Aragon (INA)-Universidad de Zaragoza (Spain), Spanish ICTS National facility. R.A. gratefully acknowledges the support from the Spanish Ministry of Economy and Competitiveness (MINECO) through project grant MAT2016-79776-P (AEI/FEDER, UE).

## Author Contributions

A.C. conceived the project, directed the study, and wrote the manuscript. L.L. carried out the synthesis, characterizations and catalytic measurements, and collaborated in writing the manuscript. D.Z. and E.A.S. carried out the in situ TEM studies with the help of L.L. R.A. performed the high-resolution STEM characterizations. P.C. carried out the in situ IR experiments. All the authors discuss the results and contribute to the formation of the manuscript.

## Additional information

**Competing interests:** The authors declare no competing financial interests.

