## [Peer Review File · Nature Communications]

Reviewers' comments:

Reviewer #1 (Remarks to the Author):

The authors presented the structural transformation of subnanometric Pt species encapsulated in zeolite crystallites under different reaction conditions by using in situ E-TEM. Overall, this is a well-written manuscript. The introduction is relevant although more references may be provided. Sufficient information about the importance of this work is described. The results are clear. The comparative experiments provide systematic studies how Pt atoms/clusters transform under various gas environment at different annealing temperatures. I recommend it to be published in nature communications after minor revision. The specific comments are listed below:

1 Electron beam irradiation: in line 91, authors stated that “very low electron beam” was used in this work. However, the imaging conditions, such as voltage and beam current used in the in situ experiments, which is critical to this study, are not described.

2, Zeolite is vulnerable to electron beam irradiation. A control experiment of imaging a same specimen at a comparable magnification as a function of time should be performed to understand whether and how the electron beam could influence the specimen, therefore helping understanding of the underlining mechanism of Pt cluster transformation. For example, does the zeolite structure changes as if so, it could contribute to the change in Pt atom dispersion?

3, nice high resolution STEM images of the MCM-22 sample after calcination in air at 550 degree were provided in the supplemental information. Similar ex-situ high resolution images that resolves the structure of Zeolites should be added in order to prove that the zeolite framework remains the same under the treatment conditions.

Reviewer #2 (Remarks to the Author):

I read through the entire manuscript with great interests. Authors directly observed the transformation of Pt subnanometer species into cluster or even particles under different atmospheres (oxidative/reductive). Also, it was interesting to observe that Pt nanoparticles and cluster can re-disperse back to subnanometric Pt species. MCM-22 pure siliceous zeolite was found to be a good support to confine the formation Pt species. The conclusions were supported by experimental results and the results were practical to give suggestion how to make real catalysts for industrial applications. I would like to highly recommend publishing the manuscript subject to answering some minor questions:

1. Why water-gas shift reaction is so unique that redispersion was not observed under high temperature, while other reactions show different results.
2. What would be the mechanism of re-dispersion at high temperature under CO+O₂ atmosphere.
3. Any clue that NO is better to re-disperse Pt cluster/particle?
4. The evidence for claiming the active site of CO oxidation seems not sufficient, why cluster plays role at low temperature and subnanometric species is important at high temperature.

Reviewer #3 (Remarks to the Author):

The manuscript reports on the evolution and stabilization of subnanometric Pt species confined in MCM-22 zeolite under reaction conditions by in situ TEM. The topic is novel and interesting, however, the data is not sufficient to support the conclusion and the manuscript are not well organized. I would suggest it to be rejected; author with a better experiment performed and good quality data may consider a resubmission. Detail comments are given in below:

1. Please provide abundant evidence to prove that the cluster in MCM-22 zeolite was subnanometric Pt. the figure S1c shows some nanoparticles already formed and it is almost same size as figure 1 b shows?
2. In Figure 1, “When the sample was exposed to H₂ (0.1 torr) at 350 oC, a few Pt clusters between 0.2-0.4 nm started to appear (see Figure 1a). When the temperature was increased to 550 oC and 700 oC in H₂ atmosphere, more subnanometric Pt species (below 0.5 nm) appeared and the average size increased (see Figure 1b and Figure 1c). The size of subnanometric Pt clusters continued to increase up to 0.5-0.8 nm when the temperature was increased to 800 oC (see Figure 1d).” How did author get the cluster size number like 0.2-0.4 nm? The image quality and magnification is not enough to have that kind of conclusion
3. Figure 1a looks like is not the same area as b, c and d show. And from e to f is more like coalesce to nanoparticles not redispersed. And we can even see the MCM-22 structure has been damaged by electron beam or heating. Which may indicate materials transformations have been triggered by heating or beam effect?
4. From figure S6, I cannot draw a conclusion that “As it can be seen in Figure S6, after exposure to NO (0.1 torr) at room temperature, the three-dimensional Pt clusters (ca. 0.8 nm) in Pt@MCM-22 sample show significant structural transformation into two-dimensional species with low contrast in STEM images.” There are no obvious changes in these images.
5. Figure S8 c and Figure S9 a are nearly the same, cannot support the conclusion of “Further increasing the temperature to 300 oC results in an almost fully redispersed sample with very few Pt nanoparticles on the surface of MCM-22 zeolite (see Figure S9).” In figure S10, I cannot see any sintering of nanoparticles too.

6. Proposed schematics in figure 4 and figure 7 cannot be supported by the experiment data~

7. STEM images are only small areas and even one nanoparticle changed in sequential images, some changes may just come from beam irradiation. The authors had come to a conclusion by these data, which is not convincing. And for in situ TEM experiment, they would better provide movies show the dynamic evolution process of Pt species.

8. There are some errors in the manuscript, such as “See Figure S3 “(Page 6). There is no label in figure S8.

Responses to Reviewers:

Reviewer #1 (Remarks to the Author):

The authors presented the structural transformation of subnanometric Pt species encapsulated in zeolite crystallites under different reaction conditions by using in situ E-TEM. Overall, this is a well-written manuscript. The introduction is relevant although more references may be provided. Sufficient information about the importance of this work is described. The results are clear. The comparative experiments provide systematic studies how Pt atoms/clusters transform under various gas environment at different annealing temperatures. I recommend it to be published in nature communications after minor revision. The specific comments are listed below:

1 Electron beam irradiation: in line 91, authors stated that “very low electron beam” was used in this work. However, the imaging conditions, such as voltage and beam current used in the in situ experiments, which is critical to this study, are not described.

Reply: Following the suggestion of the reviewer, those experimental parameters are now included in the revised manuscript (see the **Methods**). The *in situ* TEM experiments were performed on FEI Titan ETEM at 300 kV. The beam current is 0.418 nA and the dose rate is $176 \text{ e } \text{Å}^{-2} \text{ s}^{-1}$. The convergence angle, the inner angle and the outer angle is 10, 25 and 51mrad, respectively.

2, Zeolite is vulnerable to electron beam irradiation. A control experiment of imaging a same specimen at a comparable magnification as a function of time should be performed to understand whether and how the electron beam could influence the specimen, therefore helping understanding of the underlining mechanism of Pt cluster transformation. For example, does the zeolite structure changes as if so, it could contribute to the change in Pt atom dispersion?

Reply: In order to show the influences of the electron beam on the Pt@MCM-22 samples, control experiments were performed to study the influences of the beam. (see Liu, L. *et al.* Generation of subnanometric platinum with high stability during transformation of a 2D zeolite into 3D. *Nat. Mater.* **16**, 132-138 (2017).)

As shown in **Figure R1**, three high-resolution STEM images (obtained by FEI Titan 60-300) were acquired by consecutive scans. As it can be seen, after three scans, the zeolite structures have been damaged. However, the location and size of subnanometric Pt species are hardly affected. No agglomeration of Pt species was observed during the three scans. In addition, we have also checked

the stability of Pt@MCM-22 under electron beam in another TEM (JEOL 2010F). As shown in **Figure R2**, after exposure to the electron beam for nearly 8 min, we didn't observe the agglomeration of Pt species, indicating the stability of subnanometric Pt species in MCM-22 under the electron beam in the first several minutes.

It should be noted that in our work, when we performed the *in situ* TEM experiments (with Titan ETEM), we only scanned the sample after the treatments. During the various treatments, the electron beam was cut off to minimize the influences. Therefore, the evolution of Pt species, especially the changes observed in the first several acquisitions, are reliable. For instance, the agglomeration of highly dispersed Pt species to Pt clusters in CO+O₂ atmosphere are valid.

We agree with the reviewer that the electron beam can have influences on the zeolite structure and subnanometric Pt species after long exposure time, especially when working under *in situ* conditions at high temperature. Therefore, in this study, we have also performed *ex situ* studies on the Pt@MCM-22 samples and the results from *ex situ* TEM studies are consistent with the *in situ* TEM studies, indicating that the conclusions obtained from the *in situ* TEM studies are reliable.

All the above is now mentioned in the experimental section in the revised manuscript.

Figure R1. HAADF-HRSTEM images of Pt@MCM-22 acquired consecutively. The electron beam damage in the MCM-22 structure is clearly visible between (b) and (c). The destruction of the crystalline structure of MCM-22 and consequent mass loss, allow to better resolve the Pt atoms/clusters due to the change of focus conditions in (c).

Figure R2. STEM images of Pt@MCM-22 sample obtained by JEOL-2100F during the exposure to electron beam for ~8 min.

3, nice high-resolution STEM images of the MCM-22 sample after calcination in air at 550 degree were provided in the supplemental information. Similar ex-situ high resolution images that resolves the structure of Zeolites should be added in order to prove that the zeolite framework remains the same under the treatment conditions.

Reply: Following the suggestion of the reviewer, we have characterized the Pt@MCM-22 samples after different treatments by high-resolution STEM. STEM images of the Pt@MCM-22-0.17% sample, Pt@MCM-22-0.17%-400H₂ (reduced by H₂ at 400 °C), Pt@MCM-22-0.17%-200NO (after NO treatment at 200 °C) and Pt@MCM-22-0.17%-CO+O₂ (after CO+O₂ reaction) are now shown in **Figure R3** to **Figure R6**. The zeolite structures are preserved after different treatments, indicating the stability of pure-silica MCM-22 under different treatments. Furthermore, the presence of different types of Pt species are also shown in these figures. The presence of singly dispersed Pt atoms and Pt clusters are confirmed according to their intensity profiles, as shown in **Figure R3**. In the revised manuscript, these figures are also included to present more information about the samples (see the supplementary information of the revised manuscript).

We want to thank the reviewer for the very helpful comments.

Figure R3. Identification of Pt single atoms and clusters in the 0.17%Pt@MCM-22 sample. As shown in (a), Pt atoms (bright dots) and clusters (aggregates of several Pt atoms) can be seen. The intensity of a single Pt atom is displayed in (b). (c) Pt atoms and clusters in another area of the 0.17%Pt@MCM-22 sample. The intensity of a Pt cluster (~0.6 nm) is displayed in (d). (e, f) Two more representative STEM image of the 0.17%Pt@MCM-22 sample, showing the pore structure of MCM-22 and presence of subnanometric Pt species.

Figure R4. Morphological characterization of 0.17%Pt@MCM-22 sample after calcination in air at 550 °C and subsequent reduction by H₂ at 300 °C. As shown in these images, the pore structures of MCM-22 can be observed. Moreover, the presence of subnanometric Pt species in this sample is also confirmed in these images.

Figure R5. High-resolution STEM images of 0.3%Pt@MCM-22 sample after calcination in NO at 200 °C. In these images, the presence of Pt clusters and singly dispersed Pt atoms can be clearly seen. These subnanometric Pt species should come from the redispersion of Pt nanoparticles in the pristine sample after NO treatment at 200 °C.

Figure R6. Low-magnification and high-resolution STEM images of Pt@MCM-22 sample after reduction by H₂ at 400 °C. Small Pt nanoparticles around 1 nm can be observed in (a, b). However, no singly dispersed Pt atoms are observed in the high-resolution STEM images (c, d), due to the agglomeration of those highly dispersed Pt species to clusters and nanoparticles in H₂ atmosphere.

Reviewer #2 (Remarks to the Author):

I read through the entire manuscript with great interests. Authors directly observed the transformation of Pt subnanometer species into cluster or even particles under different atmospheres (oxidative/reductive). Also, it was interesting to observe that Pt nanoparticles and cluster can re-disperse back to subnanomateric Pt species. MCM-22 pure siliceous zeolite was found to be a good support to confine the formation Pt species. The conclusions were supported by experimental results and the results were practical to give suggestion how to make real catalysts for industrial applications. I would like to highly recommend publishing the manuscript subject to answering some minor questions:

1. *Why water-gas shift reaction is so unique that redispersion was not observed under high temperature, while other reactions show different results.*

Reply: Usually, to achieve redispersion of Pt species, oxidative reactants are necessary. In CO+O₂, NO+H₂ and NO+CO, O₂ or NO can serve as the oxidative reactant to trigger the redispersion of Pt species. [K. Morgan, A. Goguet, C. Hardacre, *ACS Catal.* **2015**, *5*, 3430-3445.] [R. Ouyang, J. X. Liu, W. X. Li, *J. Am. Chem. Soc.* **2013**, *135*, 1760-1771.] However, in water-gas shift (WGS) reaction, both CO and H₂O are not oxidative molecules and more importantly, H₂ is the product. The presence of H₂ in the atmosphere will cause the sintering of Pt single atoms to clusters or nanoparticles. Therefore, the redispersion behavior is not observed in WGS reaction.

2. *What would be the mechanism of re-dispersion at high temperature under CO+O₂ atmosphere?*

Reply: The interaction between Pt and CO can cause the mobility of Pt species. [R. Bliem, J. E. van der Hoeven, J. Hulva, J. Pavelec, O. Gamba, P. E. de Jongh, M. Schmid, P. Blaha, U. Diebold, G. S. Parkinson, *Proc. Natl. Acad. Sci. USA* **2016**, *113*, 8921-8926.] The authors showed that, after the exposure to CO at room temperature, Pt single atoms anchored on Fe₃O₄(001) sintered into Pt clusters. However, those Pt clusters disintegrated into Pt atoms after the evacuation at high temperature (520 K). According to DFT+U calculations, it is proposed that, the interaction between Pt and Fe₃O₄(001) surface is strong enough to stabilize the Pt atoms at high temperature. Therefore, the redispersion of Pt clusters to Pt atoms is energetically favorable at high temperature.

Nevertheless, as we mentioned in the manuscript, there are already several theoretical studies on the redispersion of metal clusters to single atoms under the CO+O₂ conditions. [Y. G. Wang, D. Mei, V. A. Glezakou, J. Li, R. Rousseau, *Nat. Commun.* **2015**, *6*, 6511.] [I. Z. Koleva, H. A. Aleksandrov, G. N. Vayssilov, *Catal. Sci. Technol.* **2017**, *7*, 734-742.] [J. C. Liu, Y. G. Wang, J. Li, *J. Am. Chem. Soc.* **2017**, *139*, 6190-6199.] Combining the experimental results from *in situ* and *ex situ* TEM studies, it is speculated that, the high temperature provides the driving force for the disintegration of Pt particles as a consequence of interaction between CO and Pt. Besides, the presence of O₂ also contribute to the stabilization of atomically Pt species since it is already known that Pt clusters can redisperse into Pt atoms in oxidative atmosphere at high temperature.

Following the comment of the reviewer, we have added more discussions and references on the evolution of Pt species in the revised manuscript (see **Page 13**).

3. Any clue that NO is better to re-disperse Pt cluster/particle?

Reply: NO is a stronger oxidant than O₂ when interacting with metal surface. Surface science studies have shown that, Pt(111) surface can be oxidized by NO at low temperature (110 K). [J. F. Zhu, M. Kinne, T. Fuhrmann, R. Denecke, H. P. Steinrück, *Surf. Sci.* **2003**, *529*, 384-396.] The dissociation of NO molecules has also been observed on the defective sites on Pt surface, leading to the formation of active O species. [W. Banholzer, *J. Catal.* **1984**, *85*, 127-134.] [Q. Ge, M. Neurock, *J. Am. Chem. Soc.* **2004**, *126*, 1551-1559.] It should also be mentioned that, the Pt(111) surface is inactive for NO dissociation due to the absence of defective sites. [R. I. Masel, *Catal. Rev.* **1986**, *28*, 335-369.] For comparison, the oxidation of Pt surface by O₂ needs higher temperature. [D. J. Miller, H. Oberg, S. Kaya, H. Sanchez Casalongue, D. Friebel, T. Anniyev, H. Ogasawara, H. Bluhm, L. G. Pettersson, A. Nilsson, *Phys. Rev. Lett.* **2011**, *107*, 195502.]

During the redispersion of metal particles, the metal particles are usually oxidized and then disintegrate as a result of the interaction between metal and molecules (such as O₂, NO, etc). [K. Morgan, A. Goguet, C. Hardacre, *ACS Catal.* **2015**, *5*, 3430-3445.] [R. Ouyang, J. X. Liu, W. X. Li, *J. Am. Chem. Soc.* **2013**, *135*, 1760-1771.] Therefore, due to the stronger interaction between Pt and NO, the redispersion of Pt is more effective during the NO treatment than O₂ treatment.

Following the comment from the reviewer, we have further discussed this issue in the revised manuscript (see **Page 10**).

4. The evidence for claiming the active site of CO oxidation seems not sufficient, why cluster plays role at low temperature and subnanometric species is important at high temperature.

Reply: In this work, we are focused on studying the evolution of subnanometric Pt species under different atmosphere and temperature. As it has been demonstrated in the manuscript, we have observed interesting structural transformation of Pt species under different reaction conditions. According to our experiment results, it is speculated that, the interaction between Pt and reactants (CO and O₂ molecules) are the driving force for the dynamic transformation of Pt species. At 100-300 °C, CO. Actually, the agglomeration of subnanometric Pt-group metal species (Pt, Pd, etc.) into nanoparticles has also been observed in some surface science systems, which can be explained by the gas-assisted Ostwald ripening mechanism. [G. S. Parkinson, Z. Novotny, G. Argentero, M. Schmid, J. Pavelec, R. Kosak, P. Blaha, U. Diebold, *Nat. Mater.* **2013**, *12*, 724-728.] [R. Bliem, J. E. van der Hoeven, J. Hulva, J. Pavelec, O. Gamba, P. E. de Jongh, M. Schmid, P. Blaha, U. Diebold, G. S. Parkinson, *Proc. Natl. Acad. Sci. USA* **2016**, *113*, 8921-8926.] When the temperature reaches 400 °C, the CO+O₂ reaction reach 100% conversion and the influence of O₂ as an oxidative reactant start to appear. At such high temperature, Pt species with smaller size may become more stable than Pt nanoparticles in CO+O₂ atmosphere, especially when Pt species are located in the constrained space in MCM-22 zeolite, leading to the redispersion of Pt nanoparticles at high temperature.

Regarding the mechanism of CO oxidation, it has been reported that the reaction mechanism is dependent on the reaction temperature. For instance, on supported Au/TiO₂ catalyst, O₂ is proved to be activated at the Au-TiO₂ interface at <80 °C while O₂ can be directly dissociated on Au nanoparticles at >80 °C. [D. Widmann, A. Krautsieder, P. Walter, A. Brückner, R. J. Behm, *ACS Catal.* **2016**, *6*, 5005-5011.] In the case of supported Pt catalysts, the reaction mechanism is also found to be dependent on the temperature. In a recent work, the kinetic behavior of Pt/TiO₂ catalyst is also dependent on the reaction temperature. [X. Yu, Y. Wang, A. Kim, Y. K. Kim, *Chem. Phys. Lett.* **2017**, *685*, 282-287.] As shown in **Figure R8**, the activation energy at >400 K is different to that at <400 K regime, implying different reaction mechanism at different temperature. According to the kinetic

studies, it is proposed that, at low temperature regime, the adsorption of CO can block the Pt surface sites, resulting in low activity for CO oxidation. At higher temperature, the poison effect due to the CO adsorption on Pt surface is no longer a problem and the reactivity increases significantly with the temperature.

Comparison of the catalytic performances of different types of Pt species (single atoms, clusters and nanoparticles) for CO oxidation are not in the scope of this work. Nevertheless, there are several recent works on supported Au catalysts, showing that Au clusters and nanoparticles are much more active than single Au atoms for low-temperature CO oxidation under the same conditions. [Wang, J.; Tan, H.; Yu, S.; Zhou, K. Morphological Effects of Gold Clusters on the Reactivity of Ceria Surface Oxygen. *ACS Catal.* **2015**, *5*, 2873-2881.] [Guo, L. W.; Du, P. P.; Fu, X. P.; Ma, C.; Zeng, J.; Si, R.; Huang, Y. Y.; Jia, C. J.; Zhang, Y. W.; Yan, C. H. Contributions of distinct gold species to catalytic reactivity for carbon monoxide oxidation. *Nat. Commun.* **2016**, *7*, 13481.]

Following the reviewer's comment, we have further discussed this issue in the revised manuscript (see **Page 13**).

We want to thank the reviewer for the very helpful comments.

Reviewer #3 (Remarks to the Author):

The manuscript reports on the evolution and stabilization of subnanometric Pt species confined in MCM-22 zeolite under reaction conditions by in situ TEM. The topic is novel and interesting; however, the data is not sufficient to support the conclusion and the manuscript are not well organized. I would suggest it to be rejected; author with a better experiment performed and good quality data may consider a resubmission. Detail comments are given in below:

1. Please provide abundant evidence to prove that the cluster in MCM-22 zeolite was subnanometric Pt. the figure S1c shows some nanoparticles already formed and it is almost same size as figure 1b shows?

Reply: In the fresh Pt@MCM-22 sample, Pt species exist as a mixture of single atoms, clusters and very few nanoparticles. In **Figure S1**, we have chosen several representative images of the fresh Pt@MCM-22 sample to show the existence of different types of Pt species.

In order to study the evolution of subnanometric Pt species during H₂ reduction treatment, we chose an area with only a few subnanometric Pt clusters when we performed the *in situ* ETEM experiments. As shown in **Figure 1a**, only a few Pt clusters can be observed in this area and it is supposed that most of Pt species in this area exist as atomically dispersed Pt. Then, the evolution of Pt species in that area under different atmosphere was studied by in situ TEM, as presented in **Figure 1**.

Following the suggestions of the reviewer, we have added more high-resolution STEM images in the revised manuscript to show the existence of different types of Pt species in the fresh Pt@MCM-22 sample, as shown in **Figure S2** in the supplementary information.

2. In Figure 1, “When the sample was exposed to H₂ (0.1 torr) at 350 °C, a few Pt clusters between 0.2-0.4 nm started to appear (see Figure 1a). When the temperature was increased to 550 °C and 700 °C in H₂ atmosphere, more subnanometric Pt species (below 0.5 nm) appeared and the average size increased (see Figure 1b and Figure 1c). The size of subnanometric Pt clusters continued to increase up to 0.5-0.8 nm when the temperature was increased to 800 °C (see Figure 1d).” How did author get the cluster size number like 0.2-0.4 nm? The image quality and magnification is not enough to have that kind of conclusion

Reply: We agree with the reviewer that it is not easy to get the accurate size of the Pt clusters. Herein, an enlarged STEM image of the Pt@MCM-22 sample after reduction by H₂ at 800 °C is presented in **Figure R7**. As it can be seen, Pt clusters with size between 0.5-0.8 nm can be clearly seen. Furthermore, some smaller Pt species can also be distinguished, which corresponding to size below 0.5 nm. Considering the configuration of the Titan ETEM (0.2 nm point resolution at 300 kV I STEM model), it is possible to see tiny Pt clusters below 0.5 nm.

As commented by the reviewer, it is difficult to determine the exact size of the Pt clusters with the Titan ETEM. Therefore, considering the reviewer's comment, to avoid confusion and to be less categorical, the description (see **Page 5** in the revised manuscript) has changed from "0.2 to 0.4 nm" to "less than 0.5 nm".

Figure R7. An enlarged STEM of Pt@MCM-22 after H₂ reduction treatment. The Pt clusters indicated by white circle is around ~1 nm, which is clearly distinguished by Titan ETEM under our experimental conditions. Therefore, the smaller Pt species indicated by red circles should be less than 0.5 nm.

3. *Figure 1a looks like is not the same area as b, c and d show. And from e to f is more like coalesce to nanoparticles not redispersed. And we can even see the MCM-22 structure has been damaged by*

electron beam or heating. Which may indicate materials transformations have been triggered by heating or beam effect?

Reply: Figure 1a is a low magnification image of this area. As it can be seen, the scale bar is also different to those in Figure 1b-Figure 1e.

The redispersion of Pt species in O₂ can be confirmed by comparing the image obtained in H₂ atmosphere at 800 °C and the image obtained in O₂ atmosphere at 500 °C. As shown here in **Figure R8**, it is clear that the number and the size of Pt species decreased after the gas was switched from H₂ to O₂. Therefore, the redispersion behavior of Pt species in O₂ is valid. Following the reviewer's comment, this is now explained in the revised manuscript (see **Page 5**).

We agreed with the reviewer that, several Pt nanoparticles were formed after treatment in O₂ at 700 °C. Those Pt nanoparticles may come from the sintering of Pt species located on the external surface of MCM-22 zeolite after long-time exposure to the electron beam at high temperature. As it can be seen in **Figure 1f**, it seems that the MCM-22 zeolite was also damaged by the beam irradiation at high temperature. Therefore, to avoid confusion, we have discussed this issue in the revised manuscript (as shown in **Page 5-6**).

Figure R8 (a) STEM image of Pt@MCM-22 sample after reduction by H₂ at 800 °C. (b) STEM image of Pt@MCM-22 sample obtained after switching the atmosphere from H₂ to O₂ at 500 °C. By

comparing (a) and (b), it is clear that the number of Pt clusters and their size decreased in O₂, indicating the redispersion of Pt species in O₂.

To show the influences of the electron beam on the Pt@MCM-22 samples, we have performed control experiments to study the influences of the beam. (see Liu, L. *et al.* Generation of subnanometric platinum with high stability during transformation of a 2D zeolite into 3D. *Nat. Mater.* **16**, 132-138 (2017).) As shown in **Figure R9**, three high-resolution STEM images (obtained by FEI Titan 60-300, working at 300 kV) were acquired by consecutive scans. As it can be seen, after three scans, the zeolite structures have been damaged. However, the location and size of subnanometric Pt species are hardly affected. No agglomeration of Pt species was observed during the three scans. In addition, we have also checked the stability of Pt@MCM-22 under electron beam in another TEM (JEOL 2010F, working at 200 kV). As shown in **Figure R10**, after exposure to the electron beam for nearly 8 min, we didn't observe the agglomeration of Pt species, indicating the stability of subnanometric Pt species in MCM-22 under the electron beam in the first several minutes.

In this study, when we performed the *in situ* TEM experiments (with Titan ETEM, working at 300 kV), we only scanned the sample after the treatments. During the various treatments, the electron beam was cut off to minimize the influences. Therefore, the evolution of Pt species, especially the changes observed in the first several acquisitions are reliable.

Figure R9. HAADF-HRSTEM images of Pt@MCM-22 acquired consecutively. The electron beam damage in the MCM-22 structure is clearly visible between (b) and (c). The destruction of the crystalline structure of MCM-22 and consequent mass loss, allow to better resolve the Pt atoms/clusters due to the change of focus conditions in (c).

Figure R10. STEM images of Pt@MCM-22 sample obtained by JEOL-2100F during the exposure to electron beam for ~8 min. In these images, the shape of the zeolite crystallite remains stable under the electron beam. Furthermore, we haven't observed significant agglomeration of Pt species in the process.

4. From figure S6, I cannot draw a conclusion that "As it can be seen in Figure S6, after exposure to NO (0.1 torr) at room temperature, the three-dimensional Pt clusters (ca. 0.8 nm) in Pt@MCM-22

sample show significant structural transformation into two-dimensional species with low contrast in STEM images.” There are no obvious changes in these images.

Reply: Following the reviewer’s suggestion, to better distinguish the structural transformation of Pt species during the NO treatment, we have introduced white squares in the images to point out the Pt species, as shown in **Figure R11** and **Figure R12**. By comparing the contrast of the Pt species, it is clear that the size of Pt clusters and nanoparticles decreased after the NO treatment. In some areas, the Pt species even disappeared after the NO treatment.

Notably, the contrast of Pt species decreased while the size increased after NO treatment. For smaller Pt species (below 1 nm), they may disappear in the STEM image after the NO treatment, indicating redispersion of Pt species. Furthermore, we have also measure the size of Pt particles after NO treatment. As presented in **Figure R13**, the size of Pt particles increases slightly after NO treatment according to the contrast profiles, implying the structural transformation of Pt particles. It has been reported that, the geometric structure of metal nanoparticles can change under reaction conditions. [M. A. Newton, *Chem. Soc. Rev.* **2008**, 37, 2644-2657.] For instance, it has been observed by *in situ* TEM that Cu nanoparticles can change from semi-spherical to planar particles when the atmosphere is changed from H₂ to mixture of H₂ and CO. [P. L. Hansen, J. B. Wagner, S. Helveg, J. R. Rostrup-Nielsen, B. S. Clausen, H. Topsoe, *Science* **2002**, 295, 2053-2055.]

Therefore, according to the *in situ* TEM results, it is proposed that, some of the Pt clusters may change from three-dimensional structure to two-dimensional structure after NO treatment, as described in **Figure R13e**. Following the comment of the reviewer, we have further discussed these results in the revised manuscript (see **Figure S7** to **Figure S9** in supplementary information) and modified the corresponding figures to make them better to be understood by the readers.

Figure R11. (a) STEM image of Pt particles in vacuum at room temperature. (b) STEM image of the same area after NO treatment at room temperature. The structural transformation of the Pt species is indicated by white rectangular. In (a), both Pt clusters and nanoparticles are present. After treatment with NO at room temperature (b), Pt clusters (<1 nm) disappear and Pt nanoparticles (>1 nm) can still be observed, although the geometric shape of those Pt nanoparticles already change.

Figure R12. (a) STEM image of Pt particles in vacuum at room temperature. (b) STEM image of the same area after NO treatment at room temperature. The structural transformation of the Pt species is indicated by white rectangular. In (a), both Pt clusters and nanoparticles are present. After treatment with NO at room temperature (b), Pt clusters (<1 nm) disappear and Pt nanoparticles (>1 nm) can still be observed, although the geometric shape of those Pt nanoparticles already change.

Figure R13. (a) STEM image of two Pt particles in vacuum. (b) STEM image of the same area after NO treatment at room temperature. (c) Intensity profile of the two Pt particles and their particle sizes. (d) Intensity profile of the two Pt particles and their particle sizes after NO treatment at room temperature. (e) Schematic illustration of structural transformation of Pt particles during the NO treatment at room temperature. Pt particles are oxidized by NO and the diameter of the particle also increased after the NO treatment.

5. Figure S8 c and Figure S9 a are nearly the same, cannot support the conclusion of “Further increasing the temperature to 300 °C results in an almost fully redispersed sample with very few Pt

nanoparticles on the surface of MCM-22 zeolite (see Figure S9).” In figure S10, I cannot see any sintering of nanoparticles too.

Reply: The NO treatment on the 0.3%Pt@MCM-22 sample aimed to improve the dispersion of Pt in MCM-22 zeolite. As shown in **Figure S10** in the revised manuscript, a large amount of Pt nanoparticles is present in the pristine 0.3%Pt@MCM-22 sample. After NO treatment at 200 °C, there are still some Pt nanoparticles observed on the surface of MCM-22 crystallites, as shown in **Figure S11a**. Furthermore, as shown in **Figure R14** and **Figure S12** in the supplementary information, single Pt atoms as well as small Pt clusters are observed in high-resolution STEM images, indicating the efficient role of NO for redispersion of Pt nanoparticles into subnanometric Pt species.

After increasing the temperature to 300 °C, most of the Pt nanoparticles disappear in the 0.3%Pt@MCM-22 sample. Only subnanometric Pt species and a few Pt nanoparticles can be observed in the 0.3%Pt@MCM-22-300NO sample, as shown in **Figure S13**. Then, the 0.3%Pt@MCM-22-300NO sample was reduced by H₂, and we didn't observe the formation of a large amount of Pt nanoparticles. The STEM images seem like 0.3%Pt@MCM-22-300NO sample. Only a few small Pt nanoparticles and Pt clusters can be observed in the STEM images (**Figure S14**), suggesting the stability of the Pt species after NO treatment.

Following the comments of the reviewer, we have further explained those STEM images in the revised manuscript (see **Page 10**).

Figure R14. High-resolution STEM images of 0.3%Pt@MCM-22 sample after calcination in NO at 200 °C. In these images, the presence of Pt clusters and singly dispersed Pt atoms can be clearly seen. These subnanometric Pt species should come from the redispersion of Pt nanoparticles in the pristine sample after NO treatment at 200 °C.

6. Proposed schematics in figure 4 and figure 7 cannot be supported by the experiment data~

Reply: In **Figure 4**, the schematic illustration shows the evolution of subnanometric Pt species under the CO+O₂ conditions. Firstly, when Pt@MCM-22 sample mainly containing atomically dispersed Pt

species is treated with CO+O₂ at 100-300 °C, those atomically dispersed Pt species will agglomerate into clusters or small nanoparticles, which has been observed by the *in situ* TEM and also the *ex situ* TEM results. When the reaction temperature is further increased to 400 °C, redispersion of Pt nanoparticles is observed, which is also supported by the TEM results as well as *in situ* IR spectroscopic results. Therefore, we have clear experimental proof on the evolution of subnanometric Pt species under CO+O₂ conditions at different temperature.

In **Figure 7**, the schematic illustration shows the evolution of subnanometric Pt species under CO+NO and H₂+NO conditions according to the *in situ* TEM results in **Figure 5** and **Figure 6**. Firstly, according to the results shown in **Figure S22** and **Figure S24**, the size of Pt clusters decreases after exposed to CO+NO and H₂+NO at 200 °C, which is probably caused by the redispersion of Pt species in the presence of NO. Further increasing the temperature to 600 °C will lead to the sintering of atomically dispersed Pt species to Pt clusters. And those Pt clusters may further grow to small nanoparticles at higher temperature (>800 °C). According to the above *in situ* TEM results, an evolution process of Pt species is proposed, as shown in **Figure 7**.

Following the comment of the reviewer, we have modified the figure captions of **Figure 4** and **Figure 7** in the manuscript.

7. STEM images are only small areas and even one nanoparticle changed in sequential images, some changes may just come from beam irradiation. The authors had come to a conclusion by these data, which is not convincing. And for in situ TEM experiment, they would better provide movies show the dynamic evolution process of Pt species.

Reply: We agree with the reviewer that the electron beam can have influences on the zeolite structure and evolution of Pt species, especially after long exposure time and when working under *in situ* conditions at high temperature. Therefore, we have also performed *ex situ* TEM studies on the Pt@MCM-22 samples after different treatments. And the results from *ex situ* TEM studies are consistent with those from *in situ* TEM studies.

The reviewer made a very good suggestion to acquire movies on the evolution of Pt species during the *in situ* TEM studies. Actually, we have tried, but unfortunately, it didn't work due to the following reasons. Firstly, the Pt@MCM-22 sample are not stable enough after long exposure to electron beam.

Considering the time for each treatment (at least 15 min), the Pt@MCM-22 structure will be destroyed after each treatment. In that case, it is impossible to study the evolution of Pt species after treatments under different conditions. Secondly, carbon contamination can occur when acquiring the STEM image in CO atmosphere. As shown in the following **Figure R15**, large amount of carbon can be observed after exposure to CO+NO at room temperature. That carbon contamination may come from the decomposition of CO under the electron beam. Therefore, when we carried out the in situ TEM experiments, the sample was treated with different gases for a certain time and then the TEM chamber was evacuated before we started to measure the sample by STEM. Based on the above reasons, we couldn't record movies to show the evolution of Pt species in this study.

Figure R15. STEM image of Pt@MCM-22 sample in vacuum and the same sample in CO+NO atmosphere.

8. *There are some errors in the manuscript, such as “See Figure S3 “(Page 6). There is no label in figure S8.*

Reply: We have checked the manuscript carefully and the errors in the figures have been corrected.

We want to thank the reviewer for the very helpful comments.

Reviewers' Comments:

Reviewer #1 (Remarks to the Author):

The authors addressed the main concerns from the reviews, the revised version of the manuscript appears to be acceptable for Nature Communications.

Reviewer #2 (Remarks to the Author):

Authors have addressed all the questions I raised. Also the points from other reviewers have been appropriately answered with necessary experiment results and new discussions and references. I would like to recommend accepting the manuscript as it is.

Reviewer #3 (Remarks to the Author):

All comments are well answered; suggest to be published after some minor revision.
There are some sentences in manuscript are confusing:

1, page 20, In summary, the dynamic structural transformation of subnanometric Pt species (including atomically dispersed Pt and Pt clusters) encapsulated in zeolite crystallites under oxidation-reduction and reaction conditions has been studied with in situ TEM.

2. in page 21 “Therefore, when studying the catalytic properties of subnanometric metal species (single atoms and clusters), in situ or operando characterization when one tries to identify the active species for specific reaction.”

Responses to Reviewers:

Reviewer #1 (Remarks to the Author):

The authors addressed the main concerns from the reviews, the revised version of the manuscript appears to be acceptable for Nature Communications.

Reply: We appreciate the positive evaluation on our manuscript.

Reviewer #2 (Remarks to the Author):

Authors have addressed all the questions I raised. Also the points from other reviewers have been appropriately answered with necessary experiment results and new discussions and references. I would like to recommend accepting the manuscript as it is.

Reply: We appreciate the positive evaluation on our manuscript.

Reviewer #3 (Remarks to the Author):

All comments are well answered; suggest to be published after some minor revision.

Reply: We appreciate the positive evaluation on our manuscript.

There are some sentences in manuscript are confusing:

1, page 20, In summary, the dynamic structural transformation of subnanometric Pt species (including atomically dispersed Pt and Pt clusters) encapsulated in zeolite crystallites under oxidation-reduction and reaction conditions has been studied with in situ TEM.

2. in page 21 "Therefore, when studying the catalytic properties of subnanometric metal species (single atoms and clusters), in situ or operando characterization when one tries to identify the active species for specific reaction."

Reply: In the revised manuscript, we have revised these sentences. Thanks for the comments.